# Body Size and Behavioural Plasticity Interact to Influence the Performance of Free-Foraging Bumble Bee Colonies

**DOI:** 10.3390/insects12030236

**Published:** 2021-03-10

**Authors:** Jacob G. Holland, Shinnosuke Nakayama, Maurizio Porfiri, Oded Nov, Guy Bloch

**Affiliations:** 1Department of Ecology, Evolution & Behaviour, Alexander Silberman Institute of Life Sciences, Hebrew University of Jerusalem, Jerusalem 91904, Israel; 2Department of Mechanical and Aerospace Engineering, New York University Tandon School of Engineering, Brooklyn, NY 11201, USA; shinn407@gmail.com (S.N.); mporfiri@nyu.edu (M.P.); 3Center for Urban Science and Progress, New York University Tandon School of Engineering, Brooklyn, NY 11201, USA; 4Department of Biomedical Engineering, New York University Tandon School of Engineering, Brooklyn, NY 11201, USA; 5Department of Technology Management and Innovation, New York University Tandon School of Engineering, Brooklyn, NY 11201, USA; onov@nyu.edu; 6The Federmann Center for the Study of Rationality, The Hebrew University of Jerusalem, Rehovot 7610001, Israel

**Keywords:** social insects, individual variability, colony composition, colony manipulation, task allocation, continuous size distribution, response threshold

## Abstract

**Simple Summary:**

Behavioural specialisation of individuals may improve the performance of groups, but could also limit the ability to switch tasks (behavioural ‘plasticity’) in response to changing group needs. In bumble bee colonies, body size, which is fixed once the bees reach adulthood, influences the tasks that bees perform, meaning that large and small bees often act as specialists. We found that when we experimentally reduced the body-size variation of colonies, some performed less well than normal. Nonetheless, in other colonies, individuals increased task specialisation or effort, which apparently compensated for the absence of large and small workers. These results suggest that both behavioural specialisation and plasticity can be important in collective group performance.

**Abstract:**

Specialisation and plasticity are important for many forms of collective behaviour, but the interplay between these factors is little understood. In insect societies, workers are often developmentally primed to specialise in different tasks, sometimes with morphological or physiological adaptations, facilitating a division of labour. Workers may also plastically switch between tasks or vary their effort. The degree to which developmentally primed specialisation limits plasticity is not clear and has not been systematically tested in ecologically relevant contexts. We addressed this question in 20 free-foraging bumble bee (*Bombus terrestris*) colonies by continually manipulating colonies to contain either a typically diverse, or a reduced (“homogeneous”), worker body size distribution while keeping the same mean body size, over two trials. Pooling both trials, diverse colonies produced a larger comb mass, an index of colony performance. The link between body size and task was further corroborated by the finding that foragers were larger than nurses even in homogeneous colonies with a very narrow body size range. However, the overall effect of size diversity stemmed mostly from one trial. In the other trial, homogeneous and diverse colonies showed comparable performance. By comparing behavioural profiles based on several thousand observations of individuals, we found evidence that workers in homogeneous colonies in this trial rescued colony performance by plastically increasing behavioural specialisation and/or individual effort, compared to same-sized individuals in diverse colonies. Our results are consistent with a benefit to colonies of large and small specialists under certain conditions, but also suggest that plasticity or effort can compensate for reduced (size-related) specialisation. Thus, we suggest that an intricate interplay between specialisation and plasticity is functionally adaptive in bumble bee colonies.

## 1. Introduction

A key organisational principle of insect societies is division of labour, whereby individuals specialise in various tasks such as caring for the brood (“nursing”), guarding the nest, or foraging for resources, meaning that they disproportionately perform these activities compared to the group average [1,2]. Division of labour is thought to improve colony-level efficiency if the costs of task switching are high, or if generalists perform tasks less efficiently than specialists [3,4,5,6]. Similar benefits are also found in other collective systems, e.g., non-insect animals [7,8], cells within multicellular organisms [9], bacterial colonies [10], and human societies [11,12]. Social insects provide outstanding model systems to unravel the organisational principles, mechanisms, and functional significance of division of labour because colony performance is assumed to have been shaped by natural selection and can be manipulated in ecologically relevant contexts [5,13].

In many insect societies, division of labour results from individuals with inherent dispositions towards (behavioural) specialisation, based on developmentally or genetically determined variation, and often accompanied by morphological, anatomical, or physiological features suited to the particular tasks they perform [2,5]. Extreme examples include size differences of more than fiftyfold in the morphological castes of ant workers [14] and developmental morphological changes in many termites [15]. These individual differences are assumed to be functionally linked to specialisation, allowing workers to be more efficient at performing certain tasks, but may come at the expense of decreasing individual behavioural plasticity, e.g., [16,17,18,19]. If these developmentally primed specialists have reduced flexibility, this may result in the need for colonies to balance the proportions of specialists, so that all tasks are performed in accordance with colony requirements. Presumably, this is why developmentally primed specialists are normally associated with large colonies, where there are enough individuals to provide a buffer against unexpected changes in the colony composition of specialists [3,14,20].

In smaller colonies, rapid changes in colony demands or disproportionate mortality of certain specialised workers could lead to a shortage in workers performing certain tasks. This may mean that retaining at least some level of plasticity is more crucial in small societies, allowing individuals to switch between different specialisms or from generalism to specialism (or vice versa). Behavioural plasticity has indeed been shown to play an important role in several social systems, and often appears to be compatible with response-threshold models [1,16,18]. In simple form, these models describe nestmates differing in internal sensory thresholds to task-related stimuli, meaning that moderately sensitive individuals will perform a task in the absence of highly sensitive individuals, via a demand-induced increase in the appropriate stimulus [1,21,22].

It is assumed that some adaptations associated with specialisation (e.g., large or small body size, and large mandibles) can limit the ability to switch tasks, but the degree to which specialisation limits plasticity is not clear [16]. Furthermore, the interplay between specialisation and plasticity has not been systematically tested in ecologically relevant contexts [23]. We addressed this issue using colonies of the bumble bee *Bombus terrestris*, which are especially well suited for studies on colony performance because of their relatively small colony size, annual cycle, and amenability to precise social manipulations in an ecologically relevant context [24,25,26,27]. Their social biology also lends itself to testing this relationship, firstly because their diverse body size range is associated with specialisation, and secondly because there is good reason to expect individual plasticity. Specifically, there is typically a behavioural continuum from the smallest to the largest workers, which tend towards specialisation in nursing or foraging, respectively [27,28,29,30,31,32,33] (for a recent review see [34]). Large body size is also associated with anatomical and morphological features which appear to contribute to increased foraging performance [35,36]. For example, large workers possess larger eyes [37,38], larger brains [39,40] and a greater density of antennal sensilla [41] which are associated with better visual and olfactory acuity. They also have more cells expressing the circadian neuropeptide Pigment-Dispersing Factor (PDF) [42] and stronger diurnal circadian rhythms [27], and they show a stronger phototactic response [43]. There is also evidence suggesting that smaller bees may be better suited to performing some in-nest activities [44]. This individual variation is largely independent of genetic effects, since bumble bee colonies are typically headed by lone singly-mated queens and so workers are closely related. Furthermore, larval developmental duration and ultimate body size are determined largely by the social environment that the brood experience during development and not by its source colony or factors in the egg [45]. Overall, this size-linked behavioural specialisation is arguably a socially complex trait [46], and could reduce individual plasticity. However, bumble bee colonies are relatively small, typically growing from ten or fewer workers in the first brood to no larger than a few hundred workers at peak size [28,47]. Thus, plasticity in worker behaviour could be important for allowing bumble bee colonies to appropriately respond to changes in the environment or in colony composition. Consistent with this prediction, individual workers have long been known to perform a variety of different tasks (e.g., [28,48]), and reducing the numbers of particular specialist workers did not reduce thermoregulation and undertaking performance in laboratory-confined colonies of *Bombus impatiens* [49].

Manipulating colony composition can be a useful approach for testing the functional significance of size-based specialisation in social insects [50], and has been used successfully in field studies with ants (e.g., [51]) and in laboratory settings with both ants and bees (e.g., [49,52,53]). Two studies in which the size distribution of mature *Bombus impatiens* colonies was manipulated under laboratory conditions, found no impact on the performance of certain tasks or on the production of worker or reproductive biomass, leading the authors to conclude that size diversity in bumble bees may be a neutral trait, rather than an adaptive one [49,53]. Here, we created colonies with a diverse or homogeneous worker body-size distribution, but with equal average size, and monitored them in free-foraging conditions to test the following hypotheses:

**Hypothesis** **1** **(H1).***Wide size variation, associated with specialisation, improves the performance of free-foraging colonies because morphologically diverse individuals better perform the tasks in which they specialise*. 

Thus, “homogeneous” colonies, in which the size diversity of workers is experimentally reduced, are predicted to perform worse than typically size-diverse colonies. 

**Hypothesis** **2** **(H2).***Homogeneous colonies attempt to compensate for the absence of developmentally primed specialists*. 

We predicted that this would be achieved by the following non-mutually exclusive mechanisms: 

**Hypothesis** **2a** **(H2a).**
*Increasing the production of small or large individuals (which are usually predisposed towards specialisation).*


**Hypothesis** **2b** **(H2b).**
*Increasing the proportion of specialists via behavioural plasticity.*


**Hypothesis** **2c** **(H2c).**
*Increasing effort in each individual (e.g., reducing inactivity, increasing foraging/nursing effort, or both).*


## 2. Methods

### 2.1. Colony Maintenance and Treatment

Incipient *B. terrestris* colonies were purchased from Pollination Services Yad Mordechai, Kibbutz Yad Mordechai, Israel. Trials 1 and 2 used different cohorts of incipient colonies (queen + up to 7 workers) and began on 14 May 2015 and 26 June 2016 (each designated ‘Day 1′), respectively. The trials were broadly similar, although several differences existed, e.g., the colonies in Trial 2, but not Trial 1, were closely genetically related to each other. Further colony rearing details are summarised in the Supplementary Methods and in Appendix A. All colonies were provided with sugar syrup and pollen and housed in wooden boxes (approx. 30 × 20 × 20 cm) from Day 2 of the experiment in one of three environmental chambers in the Bee Research Facility of the Edmond J. Safra campus of the Hebrew University of Jerusalem, Givat Ram, Jerusalem. The chambers were maintained at approximately 28 °C (mean ± SD; Trial 1 = 27.7 ± 0.7 °C; Trial 2 = 28.6 ± 0.9 °C) and 50% relative humidity (RH; mean ± SD; Trial 1 = 49.0 ± 6.7%; Trial 2 = 47.3 ± 11.6%) for the duration of experiment. Within these chambers, nests were maintained in constant darkness or dim red light for the majority of the time, with dim white light used for behavioural observations. On Day 12 or 13 (Trial 1) or Day 17 (Trial 2) of the experiment, each colony was connected to the outside environment via plastic tubing (approximately 120 cm in length), passing through the external wall of the environmental chamber and ending in a landing pad, which allowed the bees to freely forage in the campus and surrounding area. The colonies were located in close proximity to each other and to our honey bee apiary, which is known to be a challenging environment for bumble bees due to competition over resources [54]. The hot dry climate of the Jerusalem summer was also expected to be challenging for *B. terrestris*, which has a largely temperate distribution. Such challenging conditions enabled us to clearly assess the influence of treatment on colony fitness. After being connected to the outside, colonies were given small amounts of supplementary pollen and syrup, equally across colonies, until Day 22 (pollen) or 24 (syrup) in Trial 1, or throughout the experiment in Trial 2 (see Supplementary Methods for more details).

In both trials, colonies were randomly assigned to one of two treatments on Day 1: *diverse* or *homogeneous*. The initial number of workers did not differ between treatments (Mann–Whitney *U* tests: Trial 1, *U =* 15.5, *p =* 0.199; Trial 2, *U =* 12, *p =* 0.631). These treatments were then applied and maintained by continually collecting and redistributing all new workers across colonies every 1–2 days (Figure 1; total introduced workers: Trial 1, *n =* 1149; Trial 2, *n =* 2081). Specifically, homogeneous colonies received only medium workers (marginal cell length 2.4–2.7 mm), whilst diverse colonies received small (≤2.3 mm), medium, and large (≥2.8 mm) workers in similar proportions. This resulted in colonies in which the overall mean sizes of introduced workers was kept the same between treatments (Trial 1, diverse colonies = 2.59 mm, *n* = 527, homogeneous colonies = 2.59 mm, *n* = 622; Trial 2, diverse colonies = 2.55 mm, *n* = 1133, homogeneous colonies = 2.55 mm, *n* = 948), but the standard deviations differed significantly (Levene’s tests for homogeneity of variance; Trial 1, diverse colonies SD = 0.37 mm, homogeneous colonies SD = 0.14 mm, *F* = 182, *p* < 1 × 10^−15^; Trial 2, diverse colonies SD = 0.33 mm, homogeneous colonies SD = 0.11 mm, *F* = 685, *p* < 1 × 10^−15^).

On each collection day, all newly emerged workers were easily identified by their reduced yellow pigmentation and the absence of tags/marks. Upon collection, these workers were chilled on ice to assist measurement, performed using a dissecting microscope and graticule eyepiece to measure the marginal cell length of the forewings. The marginal cell length is a correlate of other body size measures and therefore appropriate for an assessment of overall body size [27,55]. The introduced workers were then individually marked by unique coloured numbered tags (the majority), or by non-unique paint marks (the minority, similar numbers between treatments). On each day, colonies in each treatment received the same number of workers, and thus individual colonies did not always receive the same number of workers that they produced. However, the number of workers introduced was adjusted to reflect the mean number of workers produced from focal colonies on each day of collection, allowing colonies to grow or shrink over time depending on the overall level of worker production. An exception to this was during Days 1–37 in Trial 1, when workers were instead introduced according to respective colony sizes to partially correct for the effects of mortality during this period (i.e., workers were added to make up colonies to equal numbers, based on the most recent census, described below); there was no significant difference between the number of workers introduced to colonies of the two treatments during these 37 days (Mann–Whitney *U* test, *U* = 13, *p* = 0.200, diverse colony mean = 80, homogenous colony mean = 68.5). The introduced workers were randomised in relation to their colony of origin—the proportion of workers from different colonies of origin was similar for the two treatments (Trial 1, *χ^2^* = 19.9, *d.f.* = 14, *p =* 0.13; Trial 2, *χ^2^* = 34.1, *d.f.* = 30, *p =* 0.28). The redistributions of workers continued throughout the experiments, until Day 54 (Trial 1) or Day 61 (Trial 2) of the experiment. The experiments ended several days later, on Day 60 (Trial 1) or Day 65 (Trial 2). At this time, all colonies showed signs of decline and most queens were dead.

### 2.2. Behavioural Observations

Starting on Day 34 (Trial 1) or Day 29 (Trial 2) and lasting until the end of the experiment, eight colonies (four per treatment) in each trial were regularly observed in 60 or 80 min sessions, which quantified in-nest and foraging behaviour of number-tagged workers in both treatments. In total, 40 in-nest sessions and 40 foraging sessions (6400 min = 106.6 h total) were conducted in Trial 1, and 52 in-nest sessions and 52 foraging sessions (7760 min = 129.3 h total) were conducted in Trial 2. This amounts to 236 h of observations in total, across both trials.

In-nest observation sessions consisted of four 4 min scans per colony, while foraging observation sessions consisted of two 20 min scans per pair of colonies. During each in-nest observation scan, each visible tagged worker was watched once for 5–20 s, and its behaviour was classified as one of the following: **tending brood* (nursing)*;* **constructing*; *grooming*; **fanning*, *feeding*, **depositing food*, *egg-inspecting*, *aggression*, *walking*, *inactive*, and, in Trial 2 only, **incubating brood*. For detailed descriptions of each behaviour, see Supplementary Methods. Of these behaviours, only those indicated with asterisks ‘*’ were considered ‘tasks’ that were later used to calculate the division of labour (see Measuring colony performance), although all in-nest behaviours were used to calculate nursing score (see below).

During foraging scans, the behaviours observed were: **leaving* = worker flying out from the end of the tube; *orientation =* leaving for orientation flight, distinguished by stereotyped slow circling flight when leaving the nest; **returning with pollen =* worker with pollen in their pollen baskets landing on the platform and entering the nest via the tube (assumed to be returning from a pollen foraging trip); **returning without pollen =* as above, but with no pollen seen (assumed to be returning from a nectar foraging trip). Events in which workers entered the tube, but flew back out without entering the colony, were not counted.

Behavioural observation data were used to compare the behaviour of workers in each of the two treatments. In order to do this, the data were first cleaned in order to correct any likely tag identification errors made during recording (for details, see Supplementary Methods). The cleaned data from both types of scans were then used to create a behavioural profile for each worker, consisting of the frequency at which she was recorded performing each behaviour over the course of the trial. Workers with fewer than five records in total (including “walking” and “inactive”) were discarded for having insufficient data, giving final sample sizes of *n =* 251 in Trial 1, and *n =* 555 in Trial 2. The remaining workers that were used for our analyses had a mean of 19 (Trial 1) or 16 (Trial 2) records per individual. These behavioural profiles were then used for several types of analysis (Figure 1). Firstly, a foraging score for each worker was calculated by summing the frequency of ‘leaving’, ‘returning without pollen’ and ‘returning with pollen’ records. A nursing score was calculated by dividing the number of brood tending records for a worker by its total number of in-nest records; this estimates the proportion of a worker’s in-nest time that was spent tending brood. There are no clear behavioural castes among bumble bee workers. Rather, a continuum ranges from individuals performing mostly nursing activities, to those performing intermediate levels of nursing and foraging, to those performing mostly foraging. Therefore, we defined thresholds to classify workers into three role classes: ‘forager’ (foraging score > 4 AND nursing score < 0.5; Trial 1, *n =* 47; Trial 2, *n =* 84), ‘nurse’ (foraging score < 2 AND nursing score > 0.4; Trial 1, *n =* 71; Trial 2, *n =* 169) or ‘intermediate’ (all other workers; Trial 1, *n =* 133; Trial 2, *n =* 302). Previous studies have shown that similar classification techniques identify ‘nurses’ and ‘foragers’ with distinct transcriptomic profiles and distinct levels of RNA editing [56,57,58].

### 2.3. Measuring Colony Performance

The following measurements were taken to assess colony performance (Figure 1). First, we recorded the total number of newly emerging workers and males in each colony (i.e., all adults produced, since no new queens were produced by the end of the experiment). Second, to estimate total adult mass, we multiplied the number of adults by their average cubed marginal cell length. Third, the number of workers per colony was periodically assessed in a colony census. Given that we controlled the number of workers introduced into the colony, we used the census as a proxy of mortality. Censuses were conducted during the evening, when all or most foragers were expected to have returned to the nest. Fourth, the number of full nectar (/syrup)-containing cells and pollen-containing cells was estimated every 1–3 days by visual inspection in the evening. These provided a measure of food collected and stored by each colony. Fifth, the data from foraging observations allowed an estimate of each colony’s foraging rate, by calculating the rate (observations per minute) of individuals leaving or returning to each nest across all observation periods. In addition to the (uniquely) number-tagged workers, the foraging observations also included workers which could not be uniquely distinguished (i.e., paint-marked workers or, rarely, unmarked workers). Sixth, the accumulated productivity of the colony, as far as maintained until the end of the experiment, was assessed by weighing each colony’s nest comb (containing wax, silk cocoons, stored food and developing brood produced by the colony). Seventh, a division of labour metric quantified the ‘division of individuals into tasks’ [59,60], which approaches one when different workers specialise on different tasks, and is robust to differences in the number of workers between colonies [61]. This was calculated based on all recorded ‘tasks’ (i.e., not all behaviours; see description of behaviours above) performed by individually-tagged workers.

### 2.4. Statistical Analyses and Hypothesis Testing

To investigate each trial (which differed in colony genotypes and the degree of genetic diversity, year and environmental conditions, supplemental food, etc.; Appendix A), the majority of analyses were performed separately for the two trials. We focussed on each hypothesis in turn.

#### 2.4.1. Comparing Colony Performance

Colony performance measures were compared between the two treatments in order to test the first hypothesis, that diverse colonies would outperform homogeneous colonies. We used Mann–Whitney *U* tests with continuity correction to compare nest comb mass and the total mass of adults produced over the whole experiment by colonies in the two treatments. Non-parametric tests were chosen for this and other per colony measures as a conservative analysis, because small sample sizes (e.g., number of colonies) made it difficult to confirm normality of data. We compared the difference in the number of workers over time between colonies in the two treatments using a linear mixed model (hereafter LMM), with response variable: Number of workers on census date (colony size); fixed predictor variables: Treatment, Day, and Treatment x Day interaction; and random intercept predictor variable: Colony ID. We compared the level of stored nectar and pollen per colony over time between the two treatments using generalised linear mixed models (hereafter GLMMs) with a Poisson error distribution and with response variable: Number of pollen cells or Number of nectar cells; fixed predictor variables: Treatment, Day, and Treatment x Day interaction; and crossed random intercept predictor variables: Colony ID and Day. For the pollen cell models, a zero-inflated model structure was used. We compared colony foraging rates between treatments using separate Mann–Whitney *U* tests for the total number of records of workers which were leaving or returning to the nest. These were calculated by dividing the total number of foraging observations (including workers not individually tagged) for each colony by the total number of minutes spent observing that colony.

#### 2.4.2. Comparing Responses to Reduced Size Distribution

We then tested the second hypothesis, which states that workers in homogeneous colonies attempt to compensate for a lack of developmentally primed specialists. Firstly, to test Hypothesis 2a, which states that homogeneous colonies increase the production of small and large workers, we compared the mean and standard deviation of worker size produced by each colony between treatments. We tested the effect of treatment within each trial using Mann–Whitney *U* tests. Secondly, to test Hypothesis 2b, i.e., whether middle-sized workers increase their level of specialisation via behavioural plasticity, we first performed Spearman rank correlation to confirm the expected overall relationship between foraging and nursing, when pooling workers from both treatments together, and also to confirm the expected relationship between size and foraging: nursing ratio in each treatment. The level of division of labour was quantified for each colony, using either medium-sized workers or all workers, and compared between treatments using Mann–Whitney *U* tests. We next compared the proportion of ‘foragers’, ‘nurses’ and ‘intermediates’ among middle-sized workers between treatments, using chi-squared tests. We also tested within medium workers in homogeneous colonies to check if they held roles in accordance with size, using one-tailed Mann–Whitney *U* tests to determine whether nurses were smaller than foragers. To test mean foraging and foraging specialisation in workers more specifically, we compared the mean and variance of per-worker foraging score between treatments using Mann–Whitney *U* tests and Levene’s tests for homogeneity of variance, respectively. The same was measured for nursing score. Thirdly, to test Hypothesis 2c, i.e., whether workers altered their levels of inactivity, we compared the mean per-worker counts of inactivity between treatments using Mann–Whitney *U* tests, using either workers of all sizes, or only medium-sized workers.

#### 2.4.3. Comparing across Trials

Lastly, to test for any consistent influence of treatment and worker behaviour on colony performance across both trials, we modelled their effect on four different performance measures of each colony. Model selection based on AICc values (Akaike’s Information Criterion corrected for small sample size) was used to select a single best model for each of the four colony performance response variables: estimated adult mass, mean number of nectar cells, mean number of pollen cells and comb mass at the end of each experiment. Given several differences between trials, we chose to always retain a predictor term for Trial in the selected model for each response, allowing the intercepts (but not coefficients) to vary between trials. Additional predictor terms used in the full models for each response were: Treatment, Mean worker inactivity, Colony Foraging Rate (both leaving and returning to nest) and Colony Foraging Rate x Treatment interaction. The model set for each response thus included the Trial predictor and all combinations of the other predictors. These model sets were chosen a priori based on their scientific credibility and the best model in each set was found by minimising AICc values [61,62].

Data processing and statistical analyses were performed using R [63]. In linear models, the presented significance of the slope of fixed covariates (e.g., ‘Day’) refers to the slope for the diverse treatment. The significance of a covariate x treatment interaction refers to the slope of the homogeneous treatment, as compared to the diverse treatment. For further details of analyses, see Supplementary Methods.

## 3. Results

### 3.1. Colony Performance

In both trials, the estimated mass of adults produced per colony did not differ between treatments (Mann–Whitney *U* tests, Trial 1, *U =* 5, *n =* 8, *p =* 0.490; Trial 2, *U =* 21, *n =* 11, *p =* 0.329; upper row in Figure 2). Colony comb mass at the end of the experiment was similar between treatments in Trial 1 (Mann–Whitney *U* test, *U =* 6.5, *n =* 8, *p =* 0.770), but was significantly higher in diverse colonies in Trial 2 (*U =* 29, *n =* 11, *p =* 0.009, lower row in Figure 2).

In Trial 1, worker number was significantly affected by treatment, with homogeneous colonies being larger (LMM, coef = 3.54, *t* = 23.9, *n* = 54 observations, *p* = 0.032; Figure 3). Day (i.e., colony age; likelihood ratio test, *χ^2^* = 0.778, *p* = 0.378), and the Treatment × Day interaction (likelihood ratio test, *χ^2^* = 3.37, *p* = 0.066) were not statistically significant. In Trial 2, worker number was significantly affected by treatment, with diverse colonies being larger (LMM, coef = −5.83, *t* = −2.23, *n* = 110, *p =* 0.047; Figure 3) and by Day, with worker number getting smaller as the experiment progressed (coef = −0.450, *t* = −8.5, *p* < 1 × 10^−12^), but not by a Treatment x Day interaction (likelihood ratio test, *χ^2^* = 0.0655, *p =* 0.798). In these LMMs from both trials, the consistency of colony random effects was low (Intraclass correlation coefficients; Trial 1 = 0 due to 0 estimated variance; Trial 2 = 0.254).

The influence of body size distribution on colony food storage also differed between trials. In Trial 1, there was no main effect of Treatment (Poisson GLMM, *z* = 0.90, *p =* 0.371), and no effect of Day on nectar cells in diverse colonies (*z* = 0.94, *p =* 0.347), but in homogeneous colonies nectar cells significantly declined more over time (with 0.03 fewer nectar cells per day, Treatment × Day interaction; *z* = −4.00, *p* < 0.001; Appendix A). In Trial 2 there was a significant increase of 0.03 cells per day in diverse colonies (Poisson GLMM, *z* = 4.38, *p* < 1 × 10^−4^) and a significant Treatment x Day interaction, with nectar cells increasing relatively less over time in homogeneous colonies (with 0.03 fewer cells per day, *z* = 4.68, *p* < 1 × 10^−5^). In regards to pollen cells, in Trial 1, there was a significant effect of Day, with 0.01 fewer pollen cells per day (zero-inflated Poisson GLMM, *z* = 2.4, *p =* 0.018; Appendix A), and a significant effect of Treatment, with 0.56 more pollen cells in homogeneous colonies (*z* = 2.8, *p =* 0.006), with no significant Day x Treatment interaction (likelihood ratio test, *χ^2^* = 1.7, *p* = 0.424). In Trial 2, there was no effect of Treatment (likelihood ratio test, *χ^2^* = 0, *p* = 1), Day (likelihood ratio test, *χ^2^* = 1.5, *p* = 0.225), nor their interaction (likelihood ratio test, *χ^2^* = 0.011, *p* = 0.919). In these GLMMs, the shared variance of random effects (adjusted Intraclass correlation coefficients [64]) were as follows: Trial 1, nectar = 0.202, pollen = 0.078; Trial 2, nectar = 0.448, pollen = 0 due to 0 estimated variance.

Colony foraging rates did not differ significantly between diverse and homogeneous colonies for either leaving the colony entrance (Mann–Whitney *U* tests; Trial 1, *U =* 3, *n =* 8, *p =* 0.200; Trial 2, *U =* 20.5, *n =* 11, *p =* 0.360), or returning to the colony entrance (Mann–Whitney *U* tests; Trial 1, *U =* 3, *n =* 8, *p =* 0.200; Trial 2, *U =* 18, *n =* 11, *p =* 0.662, Appendix A). For further details, see Supplementary Results.

### 3.2. Responses to Reduced Size Distribution

We first assessed whether colonies with reduced worker size distribution compensate by producing larger or more size-diverse workers (Hypothesis 2a). There was no effect of treatment on the mean body size of workers emerging in each colony in either trial (Trial 1, Mann–Whitney *U* test, *U =* 9, *n =* 8, *p =* 0.885; Trial 2, *U =* 14, *n =* 11, *p =* 0.931, Appendix A upper row). There was also no effect of treatment on the body size standard deviation of workers emerging in each colony in either trial (Trial 1, Mann–Whitney *U* test, *U =* 11, *n =* 8, *p =* 0.486; Trial 2, *U =* 16, *n =* 11, *p =* 0.931; Appendix A lower row).

We next tested whether homogeneous colonies increase the proportion of specialists via behavioural plasticity (Hypothesis 2b). We first confirmed the presence of a typical division of labour with a negative correlation between the frequency of nursing and foraging observations per worker (Pearson’s correlation pooling both treatments; Trial 1, r = −0.25, *n =* 251, *p <* 1 × 10^−4^; Trial 2, r = −0.10, *n =* 555, *p =* 0.013; separately per treatment, see Supplementary Results, Figure 4 upper row), and a positive correlation between body size and the foraging: nursing ratio, in the diverse treatment (Spearman rank correlation; Trial 1, rho = 0.46, *n =* 119, *p =* 1 × 10^−7^; Trial 2, rho = 0.41, *n =* 217, *p <* 1 × 10^−11^; Figure 4 lower row). We found that the overall degree of specialisation, as captured by the DoL metric (see Methods), was similar in the homogeneous and diverse colonies in both trials, when restricted to comparing medium-sized workers (Mann–Whitney *U* tests, Trial 1, *n =* 8, *U =* 6, *p =* 0.686; Trial 2, *n =* 11, *U =* 15, *p =* 1), as well as when including all workers (Mann–Whitney *U* tests, Trial 1, *n =* 8, *U =* 6, *p =* 0.686; Trial 2, *n =* 11, *U =* 17, *p =* 0.792). Consistent with this analysis, in both trials, the proportion of medium-sized workers which were ‘specialists’ (i.e., classified as nurses or foragers) was not significantly different between treatments (Chi-squared tests, Trial 1, *χ^2^* = 4.0, *d.f.* = 2, *p =* 0.135; Trial 2, *χ^2^* = 0.60, *d.f.* = 2, *p =* 0.744). Remarkably, however, in both trials, foragers were significantly larger than nurses in homogeneous colonies (one-sided Mann–Whitney *U* tests; Trial 1, *U =* 577, *n =* 61, *p =* 0.027; Trial 2, *U =* 2388, *n =* 127, *p* < 0.001; Figure 5). When further testing for changes in foraging among medium-sized workers, the mean per-worker foraging score did not differ between treatments (Mann–Whitney *U* tests, Trial 1, *U =* 2229, *n* = 168, *p =* 0.160; Trial 2, *U =* 16,057, *n =* 385, *p =* 0.270; Figure 6 upper row). However, the variance of foraging score was greater in homogeneous colonies in Trial 1 (Levene’s test, *F* = 13.2, *n* = 168, *p* < 0.001), but not in Trial 2, (Levene’s test, *F* = 0.58, *n =* 385, *p =* 0.445). The nursing count of medium-sized workers in Trial 1, was not significantly different between treatments (Mann–Whitney *U* test, *U* = 2467, *p* = 0.730; Figure 6 middle row), but the variance was slightly but significantly smaller in homogeneous colonies (Levene’s test, *F* = 5.4, *p* = 0.032). In Trial 2, both the per-worker nursing count (Mann–Whitney *U* test; *U* = 12,605, *p* = 0.011), and the variance (Levene’s test; *F* = 4.56, *p* = 0.033) were significantly higher in the homogeneous treatment among medium-sized workers.

Another form of individual plasticity relates to the time spent inactive (Hypothesis 2c). In Trial 1, the mean per-worker level of inactivity (count of observations when a worker was seen standing still, with no obvious task) was significantly lower in homogeneous colonies (Mann–Whitney *U* test, *U =* 9028, *n* = 251, *p =* 0.037; Figure 6 lower row), but there was only a non-significant trend when restricting the comparison to only middle-sized workers (*U =* 2834, *n* = 168, *p =* 0.295). Given a lack of correlation between body size and inactivity in the diverse colonies (Spearman rank correlation, rho = 0.02, *p =* 0.800), it is possible that the lack of a significant effect when restricting the comparison to middle-sized workers is due to reduced sample size and statistical power. In Trial 2, there was no significant effect of treatment when comparing the amount of inactivity for all workers in the colony (Mann–Whitney *U* test; *n =* 558, *U =* 37741, *p =* 0.802), but inactivity was higher in the homogeneous treatment when limited to only middle-sized bees (*n =* 385, *U =* 12508, *p =* 0.017; Figure 6).

### 3.3. Overall Effects on Performance across Both Trials

In pooled analyses that included data from both trials and tested the effects of various factors on colony performance using linear models, model selection identified several predictors with cross-trial effects on some of the four colony performance measures tested (Figure 7). Specifically, the best model for comb mass (adjusted R^2^ = 0.61) included an effect of *Treatment* (intercepts for the Mean Foraging Rate: for diverse treatment, Trial 1 = 78.27 g, Trial 2 = 70.76 g; for homogeneous treatment, Trial 1 = 62.69 g, Trial 2 = 55.18 g), and of *Foraging Rate* and *Foraging Rate × Treatment* interaction, with separate slopes for each treatment separately (coefficients: for diverse treatment = 7.512; for homogeneous treatment = 49.03). Thus, homogeneous colonies produce smaller comb only when their foraging rate is low. The best model for adult mass (adjusted R^2^ = 0.41) included an effect of *Foraging Rate*, without treatment interaction (intercepts: for Trial 1 = 2.918, for Trial 2 = 3.446; coefficient = 2.892). For both pollen and nectar cells, no predictors (other than trial) were included in the best model.

## 4. Discussion

Task specialisation and plasticity are both strategies that may improve collective behaviour, including in social insect colonies. However, their functional significance has been typically studied in different species (e.g., leaf cutter ants show profound morphological specialisation, and honey bees profound behavioural and physiological plasticity) and their interaction has rarely been tested by using experimental manipulations under ecologically relevant conditions. We studied the interplay between these two strategies in free-foraging bumble bee colonies under challenging environmental conditions. This ecological context forced the bees to collect resources and presumably contend with competitors, predators and parasites. Our findings provide the first evidence under field realistic conditions suggesting that both size-related specialisation and individual plasticity are functionally significant in a bumble bee species. These findings suggest that the importance of body size variability is not limited to vast colonies of ants and termites, and that plasticity in the behaviour of individual bumble bees can partially compensate for the lack of large- or small-sized workers (i.e., developmentally primed specialists) under at least some conditions.

We first tested the commonly held hypothesis that developmentally primed specialisation, as manifested in differently sized workers, improves colony performance. This hypothesis predicts compromised performance in colonies that are restricted to middle-sized, presumably less specialised, workers. In one of our trials (Trial 2), diverse colonies had significantly greater comb mass (Figure 2), more workers (Figure 3), and a relatively greater increase in number of nectar cells over time (Appendix A). Some further effects of size diversity are also apparent from the cross-trial analysis (Figure 7; see later in Discussion). These findings, which suggest an adaptive role for large and/or small workers as specialists in bumble bees, are different from two laboratory studies with food provided in small foraging arenas in which body size diversity did not affect performance in *B. impatiens* [49,53]. Our findings, however, are in accordance with the ubiquity of body size -task association reported in bumble bees (see Introduction) and also found in the current study (Figure 4). This overall effect of size diversity is also consistent with previous research showing that large workers have a range of traits that appear to make them better suited for foraging, such as larger eyes, larger brains and stronger circadian rhythms (see Introduction). The effect of reducing body size variability was, however, smaller than we expected; homogeneous colonies in Trial 1 did no worse than diverse colonies in any measure, and even had slightly, and significantly, larger colony sizes (but this difference declined towards the end of the experiment; Figure 3) and more pollen cells (Appendix A).

The second hypothesis we tested was that compensation for reduced body-size diversity would occur by plastic responses at the colony or individual level. We found no evidence for influence of worker body size diversity on the sizes of newly emerged bees reared by the colony (Hypothesis 2a). Homogenous colonies did not produce larger workers or a broader worker body size range, in either trial (Appendix A). This finding for the bumble bee is different from some ant and termite species where morphologically distinct specialised soldiers are produced in response to colony needs (e.g., [65,66]). Our findings are more consistent with plastic responses in the behaviour of individual adult bees, involving changes in both specialisation and effort (Hypotheses 2b,c), as detailed below.

A remarkable finding from our study was that, even within the limited size range of medium-sized workers in the homogeneous colonies, foragers were significantly larger than nurses in both trials (Figure 5). This appears to show that when the smallest and largest workers are lost (or in our experiments, replaced), the new smallest and largest workers are more likely to take on the roles of nurses and foragers, respectively, suggesting that biologically meaningful differences exist even between the largest and smallest of the ‘medium’ workers. This finding implies that medium-sized workers can serve as a “buffer group” because they are best positioned to plastically adjust their level of specialisation along with colony needs. In Trial 1, in which homogeneous colonies performed similarly to diverse colonies, medium-sized workers had greater variance in per-worker foraging than the same-sized workers in diverse colonies (Figure 6), consistent with increased specialisation or increased effort by specialists (Hypotheses 2b,c). In Trial 2, medium-sized workers in homogeneous colonies showed both a significantly higher level of nursing and a significantly higher variance in nursing, when compared to medium-sized workers in diverse colonies (Figure 6). Nevertheless, this increase in nursing specialisation in Trial 2 apparently did not rescue the performance of homogeneous colonies, suggesting that specialisation in foraging may be more important than specialisation in nursing. Furthermore, increased specialisation apparently cannot fully account for the improved performance of some homogeneous colonies. For example, medium-sized workers in homogeneous colonies were not more likely to be categorized as ‘nurses’ or ‘foragers’ compared to same size bees in typically diverse colonies, and the DoL metric was similar for diverse and homogeneous colonies, even when restricting the analysis to only medium-sized bees. Although our ability to measure specialisation may have been limited to some degree by the relatively low number of observations per individual, our data at least suggest that the contribution of increasing specialisation to the performance of homogeneous colonies was limited. How then were some of these colonies able to perform as well or even better than diverse colonies? In Trial 1, workers in homogeneous colonies spent significantly less time inactive compared to those in diverse colonies. Reducing the amount of inactivity has previously been suggested as a mechanism to buffer perturbations to colony composition [67,68]. We suggest that the decrease in inactivity and the increase in individual effort at least partially compensated for the lack of large and (perhaps) small specialists in homogeneous colonies, explaining the comparable performance of homogeneous and diverse colonies in Trial 1 (Hypothesis 2c).

The slight increase in specialisation and the increase in individual effort discussed above are similar to the short-term increases in foraging or brood care efforts reported for workers in bumble bee colonies in which foragers [69], or nurses [70], were experimentally removed, respectively. Such an increase in individual effort is probably associated with an increase in energy consumption and in predation risk (for the foragers) and therefore may be costly. This high cost may explain why individual plasticity could not compensate for the lack of developmentally primed specialists in Trial 2. In addition, in Trial 1, the slightly higher number of workers in homogeneous colonies significantly declined towards the end of the experiment (significant interaction between treatment and time; Figure 3), with a similar trend apparent for the number of nectar cells (Appendix A). These findings may suggest that even in Trial 1, the ability to compensate for the lack of developmentally primed specialists was limited and declined towards the end of the experiment. It is also possible that their success in the earlier stages of the experiment and its later decline relates to our compensation for mortality in the initial phase of the experiment only in this trial (arrows in Figure 3 and Appendix A).

The reason why colonies responded differently in the two trials is not clear. As detailed in the Methods and Supplementary Methods, the two trials differ in several ways (summarised in Appendix A). This includes the source colonies (that may differ genetically between the two trials), the degree of genetic variation among queens within trial, and the environmental conditions in the two years. For example, Trial 2 was conducted later into the hot and dry Jerusalem summer, when foraging may have been more difficult, with greater competition with honey bees and greater predation risk from Oriental hornets. Indeed, colony performance was lower on average in Trial 2 (adult mass and comb mass). Potentially size diversity is more advantageous in these kinds of more challenging environments (e.g., if smaller foragers suffer more than large foragers under these conditions). It is also possible that the compensation for mortality during the early part of Trial 1 may have buffered differences between treatments.

Given the differences in many responses between trials, we conducted a cross-trial analysis as an attempt to explore whether there may be consistent effects of treatment and certain behaviours on various measures of colony performance. The fit of several model terms in this analysis suggests that some of these effects may hold more generally (Figure 7). Specifically, colony foraging rate had a large positive effect on both total adult mass and on comb mass. Furthermore, for comb mass, this relationship was moderated by treatment, whereby in homogenous colonies comb mass is lower, and an increase in foraging rate contributes more strongly to comb mass. This finding is consistent with medium-sized workers being less efficient foragers than large workers, and therefore needing to increase their foraging rate to a greater extent in order to achieve a similar comb mass. The findings that no terms were selected for the pollen and nectar cell storage, may suggest that the colonies in both trials rapidly consumed the collected nectar and pollen and so were not able to translate stored food into greater success. Nonetheless, it is important to recognise the limitations of this analysis across both trials, given several differences between them, and understanding whether these effects hold more generally will require additional experiments under a more diverse range of environmental conditions. Indeed, the environmental differences between our free-foraging colonies and the lab-confined colonies in the *B. impatiens* experiments [49,53] may explain some of the different results. Additionally, the limited sample sizes of these studies, including our own (which is difficult to improve given practical considerations), make it challenging to elucidate the advantage of size-based specialisation under different environments, and this is not yet fully understood [34,71,72,73]. Nonetheless, we may speculate that, for example, the advantage of body size diversity would be more pronounced under conditions of colder and shorter days, in which large bees may be more efficient than small bees when foraging outside [38,74]. We hope that our study will encourage additional studies testing the significance of body size variability under diverse ecologically relevant conditions.

## 5. Conclusions

Our study suggests that in the same species, both specialisation (as associated with body size variation), and plasticity in the task-related behaviour of individuals can contribute to group performance in an ecologically relevant context. Such findings apparently reflect the importance of both consistency within individuals (usually viewed from a ‘division of labour’ perspective, i.e., how individuals are divided by tasks) and variation between individuals (usually viewed from a ‘task allocation’ perspective, i.e., how tasks are divided by individuals). Size diversity may provide an adaptive benefit under certain conditions, but in other cases colonies can perform fairly well when this size diversity is reduced, owing at least in part to the behavioural plasticity of individual workers. In particular, our results lend partial support to the idea that workers can increase effort and/or modify specialisation to mitigate potential losses in colony productivity when large and small workers (developmentally primed specialists) are missing. This strategy, however, may not always be sufficient to compensate for the lack of large and small workers or may be effective only for a limited period. Overall, our results, together with those of previous studies, suggest that the significance of size diversity, specialisation and flexibility are highly context dependent. These conclusions challenge the perspective that body size diversity is functionally significant only in very large colonies with morphological castes. Understanding the interplay between specialisation and plasticity, is crucial for understanding how collective behaviour is organised to appropriately respond to the various ecological conditions in which animal societies evolve and function.

## Figures and Tables

**Figure 1 insects-12-00236-f001:**
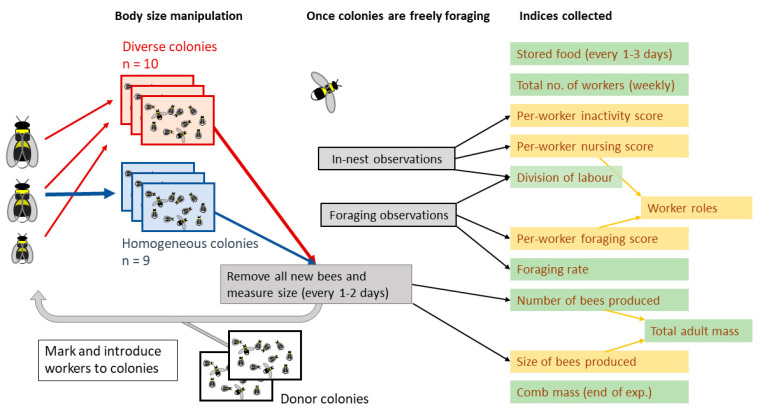
Experimental design summary. Lab colonies of *Bombus terrestris* were continuously manipulated over approximately two months, with all new workers redistributed according to body size to maintain either diverse (all sizes) or homogeneous (only medium size) distributions. After an initial establishment period, colonies were allowed to freely forage in the natural environment. Behavioural observations and other assessments were used to produce a number of individual-level indices (yellow boxes) and colony-level indices (green boxes) which were used for assessing behaviour and colony performance in downstream analyses.

**Figure 2 insects-12-00236-f002:**
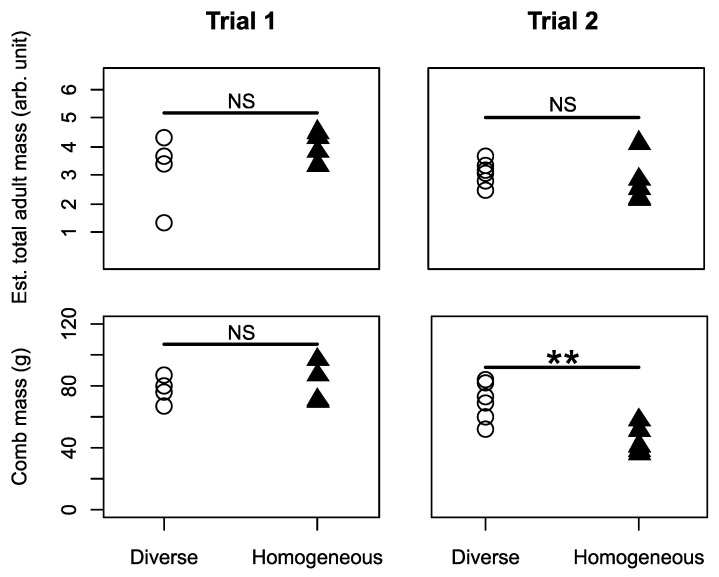
Colony performance measures as a function of worker body size distribution. Results shown for diverse or homogeneous colonies, separately for Trial 1. (left column) and Trial 2 (right column). Upper row: the total mass of adults produced over the course of the experiment per colony. Arbitrary units calculated by cubing the mean marginal cell length and multiplying it by the total number of adults produced for each colony. Lower row: nest comb mass per colony at the end of the experiment. Comb mass includes wax, silk cocoons, stored food and developing brood produced by the colony, but not adult bees. ** *p* < 0.01; NS = *p* > 0.05.

**Figure 3 insects-12-00236-f003:**
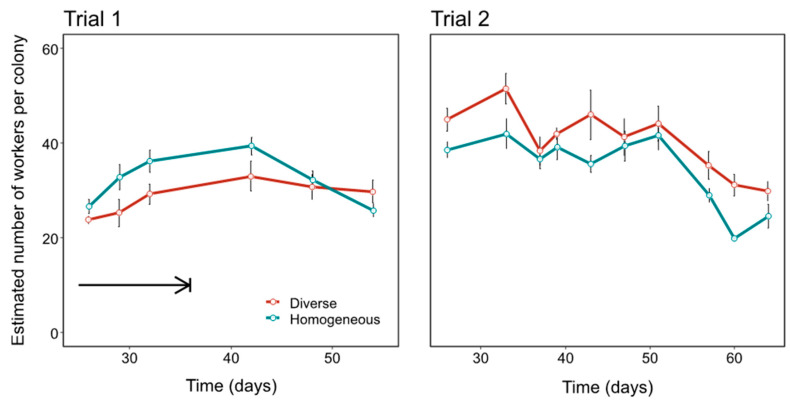
Estimated number of workers per colony over time from colony censuses. The plots show mean (±SE). Thick black arrow indicates a period in Trial 1 when workers during which introduced depending on mortality levels in an attempt to keep colony size similar between treatments at this stage (there was no corresponding period in Trial 2). Other than this period, workers were introduced in equal numbers and so differences in colony size reflect differences in worker mortality. See Methods for additional details on census procedure and statistics.

**Figure 4 insects-12-00236-f004:**
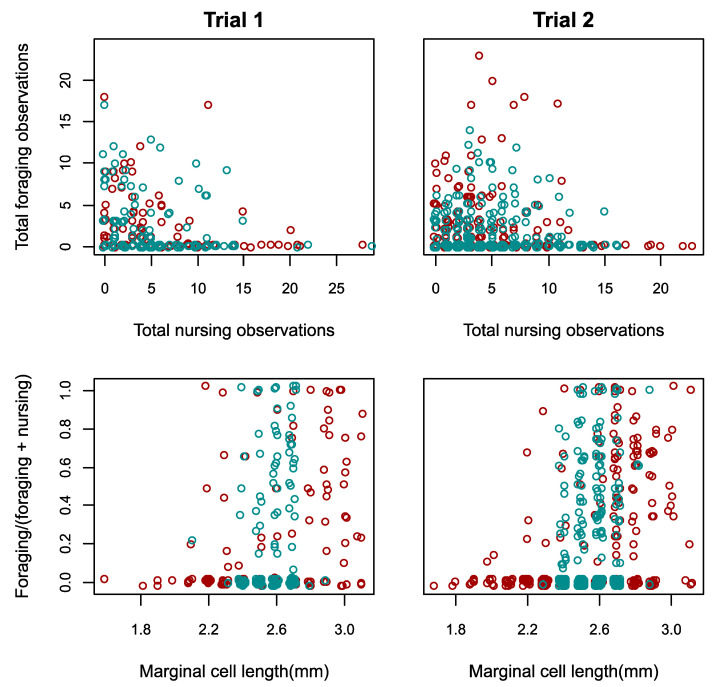
Nursing and foraging performance in size–diverse and -homogenous colonies. Red symbols—individuals in diverse colonies; blue symbols—individuals in homogeneous colonies. Trial 1—left column; Trial 2—right column. Upper row: the relationship between total foraging and nursing (brood tending) observations per worker. Lower row: foraging: nursing ratio as a function of worker body size (wing marginal cell length). In order to display multiple data points in the same position, a small amount of horizontal and vertical random noise has been added to each data point.

**Figure 5 insects-12-00236-f005:**
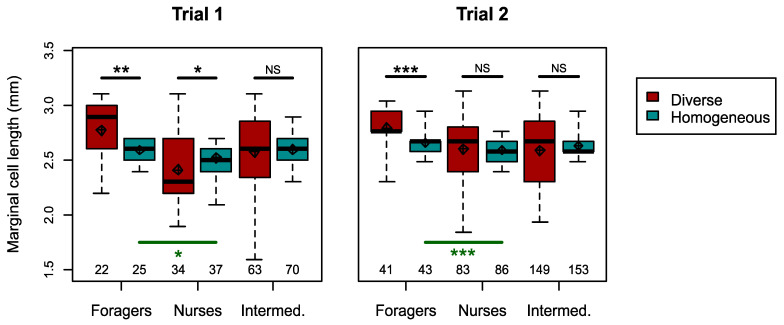
Body sizes of worker roles in size-diverse and -homogeneous colonies. Shown separately for Trial 1 (left) and Trial 2 (right). Workers categorised as foragers, nurses or intermediate, based on task activity frequencies (see Methods for details). For each trial, sample sizes of workers in each role and treatment shown at the bottom of the plot area. Diamonds = means; thick black lines = medians; boxes = interquartile ranges; dashed whiskers = ranges. Black (upper) bold lines show statistical comparisons based on individual Mann–Whitney *U* tests comparing the sizes of workers in each role between treatments. Green (lower) bold lines show statistical comparisons based on one-sided Mann–Whitney *U* tests comparing whether foragers were larger than nurses in the homogeneous treatment. *** *p* < 0.001; ** *p* < 0.01; * *p* < 0.05; NS = *p* > 0.05.

**Figure 6 insects-12-00236-f006:**
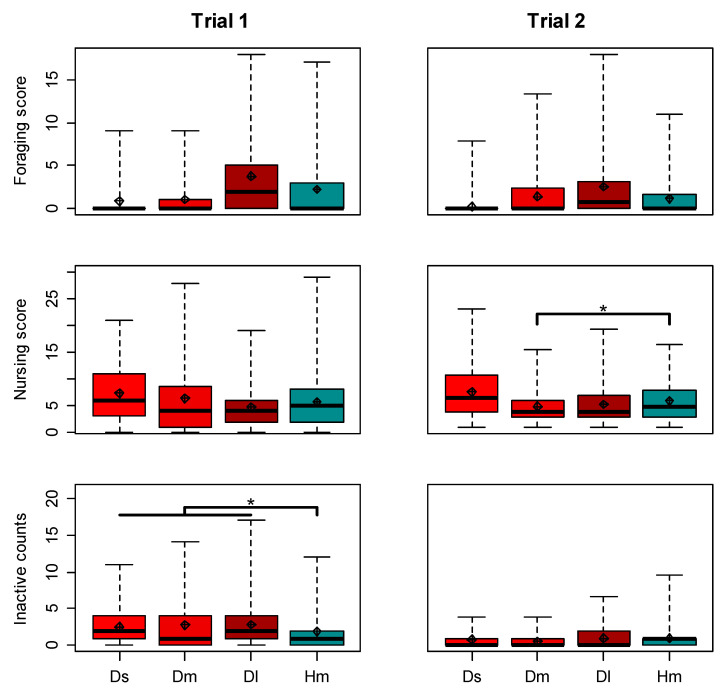
Worker foraging, nursing, and inactivity as function of body size and colony size diversity. Foraging score in upper row calculated by summing records for leaving the nest, returning to the nest without pollen and returning to the nest with pollen (see Appendix A for more details). Nursing count in middle row and Inactivity counts in lower row based on records of tending brood or standing with no obvious task, respectively. Ds = small workers from diverse colonies; Dm = medium workers from diverse colonies; Dl = large workers from diverse colonies; Hm = medium workers from homogeneous colonies. Diamonds = means; thick black lines = medians; boxes = interquartile ranges; dashed whiskers = ranges. Only significant comparisons shown, among Wilcoxon tests comparing the counts (across all workers or across only medium workers, as shown) between the two treatments. * *p* < 0.05.

**Figure 7 insects-12-00236-f007:**
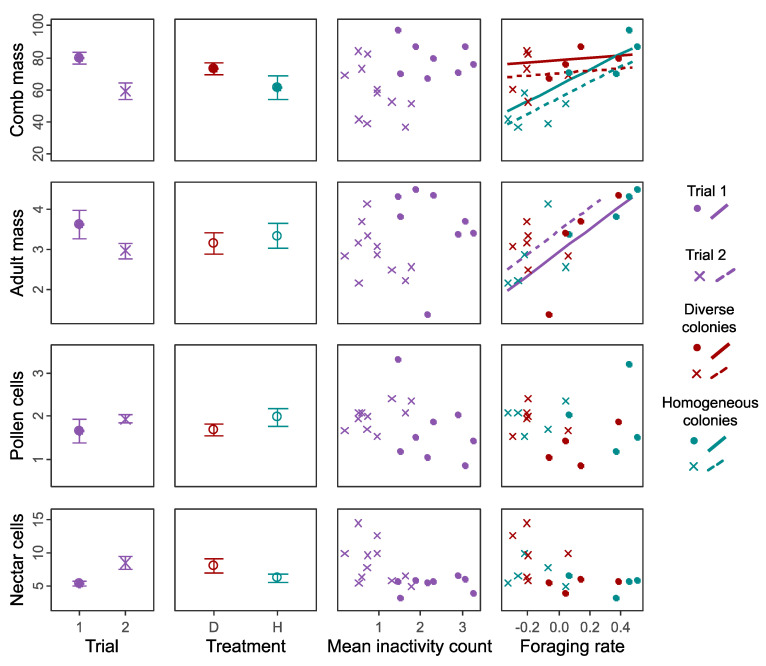
Plot matrix summary of linear models testing the effects of body size distribution and behavioural measures (matrix columns) on indices of colony performance (matrix rows). Circles or crosses represent Trial 1 or 2, respectively. Filled circles (point plots) or slopes (scatter plots) represent terms included in the best model from the model set for each response variable (selected using AICc values). Response variables were: nest comb mass at the end of the experiment (g), total adult mass (arbitrary units calculated by cubing the mean marginal cell length and multiplying it by the total number of adults produced for each colony), mean number of full pollen cells per census, and mean number of full nectar cells per census. For each response variable, the full model included the following predictor terms: *Trial*, *Treatment*, *Mean worker inactivity count*, *Colony foraging rate* (mean centred), *Colony foraging rate × Treatment* interaction. Only models with the *Trial* factor were compared, but otherwise models were selected from all combinations of predictors. For *Treatment* and *Colony foraging rate* (where an interaction term with treatment was tested), red/blue points or lines show separate data or fitted interaction slopes for the diverse/homogenous colonies, respectively, otherwise purple points/lines are used. For the fitted slopes, solid lines show the slope for Trial 1, and dashed lines for Trial 2. Point plot error bars show SEMs.

## Data Availability

The data presented in this study are openly available in researchgate.

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
