# Peer review of "Body Size and Behavioural Plasticity Interact to Influence the Performance of Free-Foraging Bumble Bee Colonies"

_insects, 2021, doi:10.3390/insects12030236_

Round 1
Reviewer 1 Report
This study investigates the influence of functional size variation and plasticity in the collective performance of Bombus terrestris colonies. It is generally well written and covers an interesting topic, providing a useful contribution to the literature. Studies in which colony trait distributions are artificially manipulated across replicates, and observed at the individual level are important, as they assist in elucidating the mechanistic basis of differing colony level outcomes, as we can see here. As such, I would recommend this work for publication, with minor revisions. Specific comments and recommendations are outlined below:
Line 53: The reference 'Deconstructing Superorganisms and Societies to Address Big Questions in Biology' (Kennedy et al. 2017), would support this statement.
Line 74: See also 'A simple threshold rule is sufficient to explain sophisticated collective decision-making' (Robinson et al. 2011).
Lines 183-184: I assume that the workers were not anesthetised in order to mark them, as they were newly emerged? It may be worth mentioning something to that effect.
Line 261: Are the boxes orange, or yellow? Considering making the colour less ambiguous, as there are some other orange boxes that appear more clearly orange.
Line 263: You may wish to subdivide the Statistical Analyses section into subheadings, to improve interpretation.
Line 279: Was model selection based on AIC/ AICc values alone? I couldn't find any information relating to absolute validation procedures in the supplementary information. If you tested model fits, it would be useful to mention the techniques employed.
Line 463: While it may not be practical based on the diversity of measures, consider adding in-figure legends to Figure 7. In at least some cases, I think that interpretation could be substantially expediated by doing so.
Lines 502-505: This is an important point, and highlights one of the difficulties in measuring colony performance.
Line 598: There is likely something to be said here in relation to conceptualising task allocation vs division of labour.
Reviewer 2 Report
In their research paper entitled “Is diversity in worker body size important for the performance of free-foraging bumble bee colonies?”, Holland et al. investigated whether the pronounced intraspecific and intra-colony body size variation in bumble bees is functionally adaptive by manipulating colonies that were able to freely forage. By measuring a variety of colony success parameters, they could demonstrate that size-variable colonies perform better than size-invariant colonies, at least in one of two trials.
The role of size polymorphism in Bombushas been much debated and many studies showed size-dependent behavioral specialization, but the adaptive significance of these observations on the colony level have, thus far, only been tested in few lab experiments. Therefore, this study, performed under realistic outdoor conditions, fills a gap in our knowledge. The presented data are novel and important for our understanding of specialization and plasticity in social insect societies. Many studies investigated specialization and plasticity and their effect in large colonies of eu-social insects (e.g. leafcutter ants, honeybees), but studies on small, primitively eusocial, societies are scarce. Understanding these will also allow for better understanding the evolution of highly specialized (behavioral or morphological) castes in eusocial societies.
The manuscript is well written and sound and I have only a few specific comments:
#1 Understanding the different outcome of Trial 1 and 2.
The authors discuss a few likely reasons for the different results in both trials. Having different starting conditions in terms of the worker numbers (i.e. replacing strategy), queen treatment, different genetic variance and different supplementary nutrition, it is a bit unfortunate, since no single reason for the differences in outcome could be pinpointed.
There is another factor that could potentially have influenced the outcome of the experiments, which is not discussed in the manuscript. The trials were conducted in two different years, one starting mid-May and the other end-June. The latter thus started about one and a half month later into the dry season with higher temperatures. Could the authors comment about the situation of flowering plants available to the bees over the course of their colony lifetime with respect to the starting date of the experiments? It may be possible that the colonies starting in May had more foraging opportunities than the ones in trial 2. Judging from Figure 2 it seems that colony success was lower for trial 2, on average, compared with trial 1. It is well possible that the adaptive function of the size polymorphism in bumblebees is not readily visible in situations where resources are not or only minimally limited (so that colonies are similarly effective), but become apparent when resources are limited and specialization really pays off (and thus colonies that are more efficient do benefit a lot more). Such difference in resource limitation between the trials is specifically mentioned as reason for the differences in supplementary nutrition between trials (Supplementary Methods).
# Assessment of Colony success
Aside from the number of workers raised and resources gathered (comb weight), one good measure for colony success would be the number of reproductive individuals raised by each colony (or the proportion of colonies that raised reproductive individuals in each trial/treatment). Counting males is problematic when using CO2-induced colonies but queens would be an option. As I understand it, no queens were raised in any of the colonies during the experiments, but it is not clear why this was the case. Where the trials too short to reach this colony phase?
# Length of the trials
It is not entirely clear how the duration of the experiment was determined. It was stopped on day 54 (T1) and day 61 (T2). Did this relate to a specific phase of the colony cycle (e.g. first queen hatched, or queen died) or arbitrarily chosen?
# Supplementary figure
In my word-file the figures are subject to some artifacts where lines of the boxes seem to have shifted to random orientations. Please check if this is the case in your file.
# Figure 2
The colored lines appear nearly indistinguishable when printed in grayscales. Could different symbols be used to allow for easy discrimination on printed copies?
L 572:“cross-trail”: I guess this should be “cross-trial”.
# References
Please go through the reference. There are many spelling errors, including author names (e.g. Refs 21, 22, 35) and the species names seem to consistently lower case and non-italicized (e.g. Refs 20, 21, 25, 28, 31, 32, 35, 39, 41-44, 50, 51, 65). There may be more, but I did not check these in more detail. The same should be done for the references in the supplementary material.
Reviewer 3 Report
This study explores the potential function of variation in the body-size with respect to the task specialization using bumble bees. The authors repeated experiments twice and found a reduced colony performance in homogeneous colonies in trial 2 but not in trial 1. By analyzing several parameters related to the colony performance and worker behavior closely, they found that increased variance of foraging score and reduced inactivity were linked to improved colony performance. Based on these results, it was concluded that the diversity in the worker body size benefits the colony while the reduced diversity can be compensated by increasing specialization or effort under some conditions.
How the individual variation influences the collective performance of group is an important and interesting issue in the biology of group-living animals. In bumble bees, the benefit of body-size variation was failed to be demonstrated in the other studies. Contrary to the earlier studies, this study not only showed the benefit in one of two trials but also partially revealed how bumblebee colonies respond and compensate reduced variation. Although the results are not very clear, the authors closely analyzed the data and succeeded to show evidences for the benefit of variation in worker body size and the existence of behavioral compensation. Their methods seem sound and manuscript is well written.
I largely agree with their conclusion but would like to request the authors to clarify a following point.
Do authors think that homogenous colonies partially compensate the reduced diversity of body size by increasing specialization in foraging in trial 1? If so, why does not increased specialization in nursing rescue the disadvantage of homogeneous body size? In trial 2, homogeneous colonies significantly increased the variance of nursing score but failed to compensate the disadvantage of homogeneity. In addition, in trial 1 in which the disadvantage of homogeneity was compensated, the homogeneous colonies even decreased the variance of nursing significantly (supplemental results, p. 10 “Nursing levels”). The authors should discuss this point because these data appear to be against their conclusion.
Minor comments:
- 165 “marginal cell”: Please give some more information for marginal cell for non-bumblebee researchers.
- 168-171: Were the SDs significantly different?
- 281-283: A factor “Day” was incorporated in the model as both fixed predictor variables and random effect. This seems to be a common method. Please give explanation.
- 413-414: The authors should explain the data of nursing score in the main text not only in supplementary materials because these results may be opposed to their conclusion.
- 474 “mean-centred”: Please give some more details for this procedure in Materials and methods (or supplementary methods).
